# Evaluation of CIN2/3 Lesion Regression in GynTect^®^ DNA Methylation-Marker-Negative Patients in a Longitudinal Study

**DOI:** 10.3390/cancers15153951

**Published:** 2023-08-03

**Authors:** Heike Hoyer, Claudia Stolte, Gerd Böhmer, Monika Hampl, Ingke Hagemann, Elisabeth Maier, Agnieszka Denecke, Christine Hirchenhain, Jan Patzke, Matthias Jentschke, Axel Gerick, Tabitha Heller, Juliane Hippe, Kristina Wunsch, Martina Schmitz, Matthias Dürst

**Affiliations:** 1Institut für Medizinische Statistik, Informatik und Datenwissenschaften, Universitätsklinikum Jena, 07743 Jena, Germany; heike.hoyer@med.uni-jena.de; 2Institut für Zytologie und Dysplasie (IZD), 30159 Hannover, Germany; stolte@izd-hannover.de (C.S.); boehmer@izd-hannover.de (G.B.); 3Frauenklinik, Universitätsklinikum Düsseldorf, 40225 Düsseldorf, Germany; hampl@med.uni-duesseldorf.de; 4Abts+Partner Partnerschaftsgesellschaft, 24103 Kiel, Germany; i.hagemann@abts-partner.de; 5Praxis Dr. Elisabeth Maier, 80796 München, Germany; info@frauenarztpraxis-maier.de; 6Klinikum Wolfsburg, 38440 Wolfsburg, Germany; denecke.agnieska@mh-hannover.de; 7Klinik und Poliklinik für Frauenheilkunde und Geburtshilfe, Technische Universität Dresden, 01307 Dresden, Germany; christine.hirchenhain@uniklinikum-dresden.de; 8CytoConcept, 44145 Dortmund, Germany; patzke@cytoconcept.de; 9Klinik für Frauenheilkunde und Geburtshilfe, Medizinische Hochschule Hannover (MHH), 30625 Hannover, Germany; jentschke.matthias@mh-hannover.de; 10Praxis Dr. Axel Gerick, 52072 Aachen, Germany; praxis@drgerick.de; 11Zentrum für Klinische Studien (ZKS), Universitätsklinikum Jena, 07747 Jena, Germany; tabitha.heller@med.uni-jena.de; 12Ongnostics GmbH, 07749 Jena, Germany; juliane.hippe@oncgnostics.com (J.H.); kristina.wunsch@oncgnostics.com (K.W.); martina.schmitz@oncgnostics.com (M.S.); 13Klinik und Poliklinik für Frauenheilkunde und Fortpflanzungsmedizin, Universitätsklinikum Jena, 07747 Jena, Germany

**Keywords:** cervical precancers, CIN, methylation markers, regression, predictive markers, watchful waiting

## Abstract

**Simple Summary:**

Precancerous cervical lesions, especially among young women, are frequently cleared by the immune system. Consequently, immediate surgical treatment is not mandatory and patients are monitored closely for up to 24 months. To avoid anxiety and to allow for a more individualized treatment, molecular markers that could predict the outcome of the lesion would be desirable. Women with a high risk of progression could be identified earlier. In this study, we could show that the majority of lesions that were negative for the cancer-associated methylation markers comprising the test GynTect^®^ have regressed over time.

**Abstract:**

Cervical intraepithelial neoplasia (CIN) grade 2/3 has a high spontaneous regression rate, especially among women ≤29 years of age. To reduce overtreatment, reliable prognostic biomarkers would be helpful. The main aim of this study was to analyze the negative predictive value of the methylation marker panel GynTect^®^ for lesion regression. In this prospective, multicenter, longitudinal observational proof-of-concept study, women aged ≤29 years with histologically confirmed CIN2 (n = 24) or CIN3 (n = 36) were closely monitored without treatment for up to 24 or 12 months, respectively. The outcome was either regression, persistence, or progression of the lesion. For each patient, a single baseline sample (V0) for cytology, hrHPV detection and methylation analysis was taken. In a primary analysis, the negative predictive value (NPV) of a GynTect^®^-negative test result at V0 for regression was determined. We tested the null hypothesis NPV ≤ 70% against the alternative hypothesis NPV ≥ 90%. Twelve of the eighteen GynTect^®^-negative CIN2 patients showed regression (NPV = 67%, 90% CI 44–85%, *p* = 0.53). Of the 27 GynTect^®^-negative CIN3 lesions, 15 regressed (NPV = 56%, 90% CI 38–72%, *p* = 0.92). Although the majority of GynTect^®^-negative lesions regressed, the postulated NPV of ≥90% was not observed. Thus, the clinical relevance for an implementation of the GynTect^®^ assay for patients undergoing watchful waiting remains questionable. Further studies with longer observation periods should be undertaken.

## 1. Introduction

Primary screening for cervical cancer and its precursors based on HPV-DNA or -RNA testing has been introduced in most European countries [1,2]. An HPV-positive result requires a triage test in order to identify women in need of colposcopic assessment. Cytology is one of the commonly used triage instruments, either as Pap-stain or in the form of an immunostain, particularly for p16/Ki67 overexpression [3,4,5]. Another well-explored triage approach is the use of methylation markers [6]. Several CE-certified methylation-marker-based diagnostic tests for cervical cancer screening are commercially available [7]. The clinical utility of diagnostic methylation markers relies on the fact that methylation is characteristic of cancer cells and their precursors, and increased levels are correlated with disease development [8,9]. Although highly sensitive and specific to the detection of cervical intraepithelial neoplasia grade 2 or 3 (CIN2/3) and cervical cancer, no methylation marker (or combination thereof) has been identified that detects all CIN2/3 to date. A possible explanation for this apparent weakness is the observation that not all CIN2/3 lesions have the potential to advance to cervical cancer. Especially among young women, these lesions frequently regress to normal [10,11,12,13]. Indeed, these circumstances are the basis for the current concept of watchful waiting, which is standard care in several countries for young women with CIN2/3 [14,15]. It may thus be argued that methylation markers primarily identify those lesions with the potential to progress. Conversely, methylation-negative lesions are likely to regress. However, direct proof provided by prospective longitudinal studies is, thus far, limited [16,17]. 

The epigenetic-marker-based cervical cancer diagnostic test GynTect^®^ detects highly methylated promoter regions of the human genes ASTN1, DLX1, ITGA4, RXFP3, SOX17, and ZNF671 [18]. The test has been shown to have a high sensitivity for CIN3+ and has detected all carcinoma cases in all studies performed to date. The detection rates observed show a clear correlation with the severity of the underlying histopathology grades [6,9,19]. 

In order to address the usefulness of the GynTect^®^ assay as a predictor for lesion regression, we initiated a prospective, longitudinal, and multi-centered proof-of-concept study in Germany, termed GynTect-Pro. We observed a cohort of women with histology-confirmed CIN2 or CIN3 undergoing conservative management, also referred to as watchful waiting. We postulated a negative predictive value (NPV) of ≥90% among GynTect^®^-negative women. The study tested the null hypothesis NPV ≤ 70% against the alternative hypothesis NPV ≥ 90%.

## 2. Materials and Methods

### 2.1. Study Population and Sample Collection

This prospective, longitudinal, and multi-centered observational study was registered at the German Clinical Trials Register on 2 September 2017 (DRKS-ID: DRKS000125771). The study was conducted in a setting in which young women with CIN2/3 were closely monitored without primary surgical intervention. Overall, 9 study centers in Germany (3 University Hospitals and 6 gynecologists in private practice) were involved. Women with clinical suspicion of CIN2/3 lesions were invited for colposcopy and, if indicated, a biopsy was taken. Women aged from ≥18 to ≤29 with histopathology confirmed CIN2 or CIN3 were asked to participate in the study. Exclusion criteria were women with lesions where the entire squamous-glandular junction was not visible, women with glandular atypia, cases in which an invasive component could not be excluded, and pregnancy. After inclusion, one baseline cervical sample (V0) was taken and transferred to ThinPrep PreservCyt Solution (Hologic, Marlborough, MA, USA). This sample was used for cytology, partial hrHPV genotyping and methylation analyses.

Participants were prospectively monitored every 6 months for a period of 12 months (CIN3) and 24 months (CIN2) according to the German guidelines for watchful waiting [20]. Moreover, at each follow-up visit, a colposcopic examination was perfomed and, if indicated, a biopsy was taken. In cases in which the transformation zone was not completely visible, the study was terminated. The gynecologists were blinded to the outcome of the methylation analyses. Disease status at the timepoint of follow-up visits and at the last visit was classified as regression, persistence or progression, based either on clinical judgement (cytology, colposcopic impression), biopsy or conization. In the case of CIN2 and CIN3 at baseline, regression was defined as ≤CIN1 and ≤CIN2, respectively. Persistence was defined as an unaltered status compared to baseline, whereas, for CIN2, progression was defined as ≥CIN3 and, for CIN3, as an increase in colposcopic volume, adenocarcinoma in situ or carcinoma.

Study data were recorded via the customized electronic case report forms applied by the Center of Clinical Studies at Jena University Hospital (JUH). The database comprised all clinical data (including cytology and HPV testing) collected by the different study sites. Methylation data were recorded separately by the laboratory staff, who were blinded to all clinical data. All data were managed and merged by JUH staff, who were also responsible for quality control, plausibility, and study monitoring. The study was approved by the institutional review board at JUH (vote: 5166-05/17) for the principal investigator and by the local review committees for all participating sites, respectively.

### 2.2. Cytology, HPV Testing and Methylation Analyses

Liquid-based cytology and HPV testing of all samples were conducted at the IZD in Hannover. Cytology was classified according to Munich 3 nomenclature. For HPV testing, the cobas^®^ HPV assay (Roche, Mannheim, Germany) detecting HPV16 and 18, as well as 12 further high-risk (hr) HPV types, was used. All methylation analyses (GynTect^®^ assay) were conducted by oncgnostics GmbH, Jena. For this purpose, surplus samples were sent from Hannover to Jena. Molecular testing was performed blinded to cytology and histology outcomes. The GynTect^®^ assay was performed according to the instructions for use. Samples were treated with bisulfite using the EpiTect^®^ Fast DNA Bisulfite Kit (Qiagen, Hilden, Germany) to allow for detection of the methylated DNA marker regions in the subsequent GynTect^®^ realtime PCR assay. In brief, LBC samples were vortexed for a few seconds, and 1 mL of each sample was transferred into a 2 mL Eppendorf tube for use in the assay. Cells were centrifuged, 900 µL of the supernatant was removed, and the pellet was resuspended in the remaining ca 100 μL supernatant. A total of 40 μL of the resuspended cells were directly used for the chemical bisulfite treatment, as previously described [21]. After elution of the bisulfite-converted DNA with 20 μL water, 70 μL water was added and 10 μL was applied to each reaction in the GynTect^®^ real-time methylation-specific PCR (qMSP) assay [9,19]. The qMSPs were run on a 7500 Real-Time PCR system (Applied Biosystems, Thermo Scientific, Waltham, MA, USA). For each marker, the Ct-value was determined and a delta Ct was calculated between the Ct-value of the quality control marker ACHE and the Ct-value for each marker. To be scored as positive, the delta Ct has to be ≤8 for ASTN1, ≤9 for DLX1, ITGA4, RXFP3, SOX17 and ≤10 for ZNF671. Each marker has a score, if positive (DLX1 score 1; ASTN1, ITGA4, RXFP3, SOX17, each score 2; ZNF671 score 6), and GynTect^®^ is considered to be positive if the total GynTect^®^ score is equal to or higher than 6. Samples were valid if the Ct value for the second control marker, ACHE, was ≤32 for the respective sample [9].

### 2.3. Sample Size Calculation and Statistical Analysis

The primary study endpoint was clinical regression. We postulated that regression among methylation-negative women with CIN2/3 would have to be ≥90% to be of clinical relevance in the intended setting of watchful waiting. This assumption is based on reported regression rates of up to 70% in this age group, irrespective of the methylation status [10,11]. For our proof-of-concept study, we used a single-stage phase II design. The negative predictive value (NPV) for the null hypothesis would have to be ≤70% (NPV_0_) and, for the alternative hypothesis, ≥90% (NPV_1_). On this basis, and by considering a one-sided significance level of 0.05 and a power of 80%, the sample size would have to be 28 GynTect-negative women for the CIN2 and CIN3 group [22]. Considering the methylation rates observed for this age group, in our previous studies [9,19], at least 30 patients with CIN2 and 44 patients with CIN3 needed to be enrolled. We estimated the two-sided 90% exact confidence interval (CI) for NPV, assuming a binomial distribution. The above hypotheses were tested by the exact binomial test and a one-sided significance level of 0.05. The null hypothesis NPV ≤ 70% was rejected if the one-sided p value was smaller than 0.05. Post hoc, we explored the association between the methylation status and cytology and HPV type, respectively, at baseline, using Fisher’s exact test.

## 3. Results

From December 2017 to February 2020, 109 women with suspected CIN2 or CIN3 lesions were recruited. For 32 women, the suspected diagnosis could not be confirmed by histopathology. Furthermore, 17 women had to be excluded for other reasons. Overall, 24 women diagnosed with CIN2 and 36 women with CIN3 were available for evaluation (Figure 1). The medium age of these women was 22 (range, 20–29 years) and 22.5 (range, 18–25 years) for CIN2 and CIN3, respectively. Further patient characteristics at baseline are shown in Appendix A. Participants were monitored prospectively every 6 months for a period of up to 24 months. The assessment of the disease status at the end of the study was either based on histology from conization (n = 15) or biopsy (n = 36) or clinical judgement via colposcopy (n = 9) (Figure 1). Median follow-up time was one year (ranging from 0.2 to 1.4 years) for CIN3 patients and 1.8 years (ranging from 0.5 to 2.6 years) for CIN2 patients. Nine CIN3 and 12 CIN2 patients finished the study prematurely for the following reasons: surgical intervention for safety reasons (n = 7), patients’ concern (n = 2) and loss of contact (n = 12).

Clinical regression occurred in 17 of 24 (71%) women with CIN2 and in 20 of 36 (56%) women with CIN3. Clinical persistence was seen for 6 of 24 (25%) women with CIN2 and for 16 of 36 (44%) women with CIN3. Only one patient with CIN2 showed progression. 

In total, 18 of 24 CIN2 and 27 of 36 CIN3 patients had a negative GynTect^®^ result at the study entry point. Over time, 12 of 18 GynTect^®^-negative CIN2 patients showed regression (NPV = 67%, 90% CI 44–85%, *p* one-sided = 0.53), whereas 5 persisted during the follow-up period (2 of them with follow-up <1.5 years). One patient showed progression (Figure 1 and Figure 2). Of the 27 GynTect^®^-negative CIN3 lesions, 15 regressed (NPV = 56%, 90% CI 38–72%, *p* one-sided = 0.92) and 12 showed persistence (5 of them with follow-up <0.8 years). None of the women were diagnosed with cervical cancer. Four CIN2 and six CIN3 were tested as GynTect^®^-positive, and two CIN2 and three CIN3 showed an invalid result at the entry point (Figure 1 and Figure 2).

The primary aim of this study was to evaluate the regression of CIN2/3 lesions after an observation period of one year (CIN3) and 2 years (CIN2) for patients with a negative GynTect^®^ result. As mentioned above, several patients terminated the study prematurely. Thus, in cases in which the last data point was persistence, the probability of final regression would be underestimated, whereas that of final persistence/progression would be overestimated. Therefore, a sensitivity analysis was performed in which we assumed regression if women terminated the study prematurely, at CIN2 < 1.5 and CIN3 < 0.8 years of follow-up, with the status persistence. Under these premises, the negative predictive value of a GynTect^®^ test taken at baseline would be 78% (14 of 18) (90% CI 56–92%, *p* one-sided = 0.33) for women with CIN2, and 74% (20 of 27) (90% CI 56–87%, *p* one-sided = 0.41) for women with CIN3. Despite these considerations, the postulated NPV of ≥90% (null hypothesis NPV ≤ 70%) for a GynTect^®^-negative result could not be proven.

The secondary aim of the study relates the positive methylation status with disease persistence or progression. Since the number of GynTect^®^-positive patients was very small, we omitted the estimation of the positive predictive value. Persistence was observed in one of four CIN2 patients and four of six CIN3 patients (Figure 1 and Appendix A).

In post hoc analyses, we also explored the relationship between the methylation status and cytology and HPV type, respectively, at baseline. Although, among the HSIL group, the number of GynTect^®^-positive cases was higher in comparison to the ≤LSIL group, 23% (8 of 35) vs. 8% (1 of 13), the difference was not significant (Appendix A). An interesting association was found by comparing HPV type with the methylation status: HPV16- or HPV18-positive cases (35%, 8 of 23) were significantly more frequently methylated than cases with other hrHPV types (4%, 1 of 25), *p* < 0.01 (Appendix A).

## 4. Discussion

This prospective, longitudinal, and multi-centered cohort study was conducted in a setting of watchful waiting, which allows for the conservative management of women with CIN2/3 lesions. This option is especially attractive for young women (≤30 years) who wish to avoid overtreatment and prevent obstetric complications. Of the 60 women with CIN2 or CIN3, 43 women did not receive surgical treatment at the end of the study period of up to 24 months (CIN2) and 12 months (CIN3). Overall, clinical regression was observed for 17 of 24 (71%) and 20 of 36 (56%) women with CIN2 and CIN3, respectively. In a similar study conducted by Kremer and colleagues, regression was evident in 67 of 114 (59%) CIN2/3 cases within an observation period of 24 months [16]. In their study, the regression rate may be somewhat lower because of the higher median age of their cohort (30, range 20–53 years vs. 20, range 20–29 years).

Unexpectedly, the proportion of methylation-negative test results among women showing regression was lower than we postulated: only 12 of 18 (67%) and 15 of 27 (56%) CIN2 and CIN3 regressed, respectively. Thus, the postulated NPV of ≥90% could not be proven. The postulate was based on a previous cross-sectional study, in which 73.2% (95% CI 59.7–84.2%) of women (<30 years) with CIN3 were GynTect^®^-negative [19]. Indeed, in the present study, 75% (27 of 36) of CIN3 were GynTect^®^-negative at baseline, which underlines our previous results. In the above-mentioned recent study by Kremer and colleagues, the clinical regression of CIN2/3 was shown to be associated with the absence of methylation markers FAM19A4/miR124-2. In their study, regression was seen in 74.7% and 51.4% of women with CIN2/3 who were methylation-negative and methylation-positive, respectively, at baseline (*p* = 0.013). By combining CIN2 and CIN3, the regression rates observed in our study were similar, i.e., 60% (27/45) and 50% (5/10) for GynTect^®^-negative and GynTect^®^-positive women, respectively, at baseline.

In addition to methylation, numerous other factors were shown to be prognostic for the spontaneous regression of high-grade lesions, particularly that of CIN2. Koeneman and colleagues have shown that not smoking and nulliparity were significantly associated with disease regression [23]. Even the vaginal microbiota is associated with CIN2 regression [24]. Moreover, further predictive factors to be considered for the conservative management of CIN2 are the history of previous cytology findings and the HPV genotype [25]. Also, transcriptome analyses have provided evidence that gene signatures may be useful for the identification of CIN3 with the potential to regress [26].

The main weakness of our study is the high number of cases with persistence and the virtual lack of progression. Moreover, since the follow-up of some patients was shortened, the probability of a final regression could be underestimated and that of final persistence overestimated. We addressed this problem by a sensitivity analysis, where we assumed regression for some patients with lesion persistence and incomplete follow-up. A longer observation period, especially for women with CIN3, might have been more favorable for the outcome of the study, but this would have meant a deviation from the current German guidelines, according to which persistent CIN3 has to be treated after 12 months of observation. Another important aspect to be considered is that regression of a lesion may be induced by the repeated biopsies taken for safety reasons during follow-up. Moreover, the calculated sample size of 30 for CIN2 and 44 for CIN3 was not reached. Additionally, for the disease status of 9 of 60 patients at the end of the observation period, only clinical data and no histopathology were available. Finally, each study center provided data from their local pathologist. Reevaluation by two expert pathologists of the only case which showed progression in fact revealed CIN2 persistence. The study may have profited from a consensus pathology for all cases.

In another recent prospective study conducted by Zhang and colleagues, the triage performance and predictive value of the GynTect^®^ assay was analyzed for a cohort comprising 1758 hrHPV-positive women who received a GynTect^®^ test at trial inclusion. Satisfactory risk stratification to detect cervical intraepithelial neoplasia of grade 2 or worse (CIN2+) was demonstrated by the methylation panel with an odds ratio (OR) of 11.3 among methylation-positive women compared to their methylation-negative counterparts. Triaging with the methylation panel would have reduced the colposcopy referral rate by 67.2% with a sensitivity and specificity of 83.0% and 69.9% to detect CIN2+. Moreover, the substantially lower risk of CIN2+ among triage-negative women over 3 years underlines the predictive value of the GynTect^®^ methylation markers [17].

With respect to our post hoc analyses, the observed increase in GynTect^®^ positivity rates with the increasing severity of the cytological findings are in line with our previous studies, which were primarily based on histological findings [9,18]. Moreover, the observation that the HPV16/18-positive cases of the current study are significantly more frequently methylated than the non-16/18 hrHPV cases is highly interesting and may reflect the cancer progression risk of the underlying lesion. This would also be in keeping with the fact that, among the hrHPV types, HPV16 is known to confer the highest risk for cervical cancer [27,28].

## 5. Conclusions

The hypothesis that ≥90% of CIN2/3 with a negative GynTect^®^ test at baseline would regress within an observation period of 24 months (CIN2) or 12 months (CIN3) could not be proven. Instead, an NPV of 67% for CIN2 and 56% for CIN3 was observed. Since follow-up of some patients was shortened, the probability of a final regression could be underestimated, and that of a final persistence overestimated. On the other hand, lesion regression may be induced by the repeated biopsies taken for safety reasons during follow-up. Thus, the clinical relevance for the implementation of the GynTect^®^ assay for patients undergoing watchful waiting remains questionable. For valid NPV estimates, we need to study the natural development of CIN2/3 lesions over a more relevant period, a task which is difficult to implement in the targeted clinical setting.

## Figures and Tables

**Figure 1 cancers-15-03951-f001:**
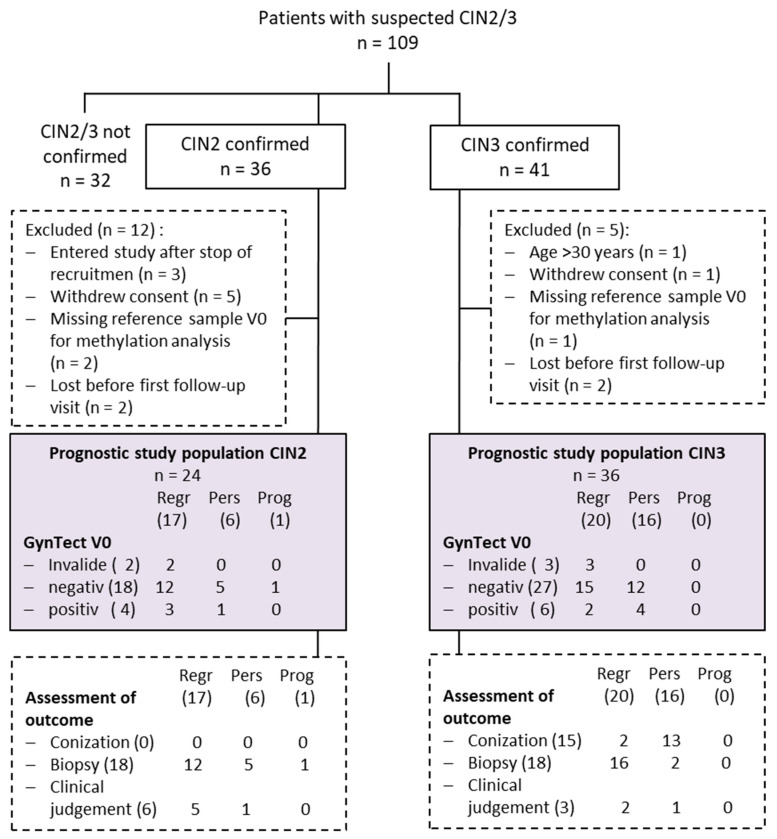
Flow diagram of the study population and outcome. Abbr.: CIN—Cervical Intraepithelial Neoplasia, Regr—regression, Pers—persistence, Prog—progression.

**Figure 2 cancers-15-03951-f002:**
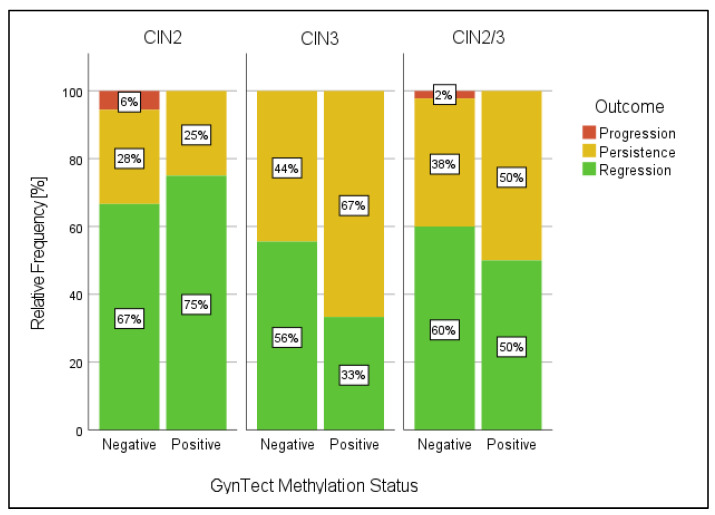
Relationship between GynTect^®^ methylation status at baseline (V0—valid results) and outcome in patients according to the grading of lesion at study entry: CIN2 (18 methylation negatives, 4 methylation positives), CIN3 (27 methylation negatives, 6 methylation positives), CIN2/3 (45 methylation negatives, 10 methylation positives). Percentages displayed in bars may not add up to 100% due to rounding. Complete data are shown in Appendix A.

## Data Availability

All data generated in this study are available from the corresponding author on reasonable request. Privacy and ethical restrictions will be accounted for.

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
