# Peer review of "Evaluation of CIN2/3 Lesion Regression in GynTect® DNA Methylation-Marker-Negative Patients in a Longitudinal Study"

_cancers, 2023, doi:10.3390/cancers15153951_

Round 1

Reviewer 1 Report

This is an interesting and important study that investigates a specific methylation assay as a possible clinical marker for CIN 2+ regression. The study design is good. However, the study cohort is small, and one possibility would be to wait a few more years to obtain a larger study cohort before publishing and/or gather results from different hospitals (with the same watchful waiting strategy).

The simple abstract should be improved. It is not just decrease in anxiety that will be reduced by clinical implementation of biomarker tests for regression. Treatment could also be more individualised with improved prognostication. Women with high risk of progression could be identified earlier.

A table summarising outcomes for patients with negative vs positive GynTect test in CIN2, CIN3 and CIN2 and 3 combined would be clarifying.

A discussion of other studies investigating potential markers for CIN regression should be included (e.g. PMID: 34830895; PMID: 34226019; PMID: 32332850¸ PMID: 31055567; PMID: 31079058).

The quality of English language is satisfactory. However, the punctuation should be improved.

Author Response

Thank you for the valuable comments!

Query 1: This is an interesting and important study that investigates a specific methylation assay as a possible clinical marker for CIN 2+ regression. The study design is good. However, the study cohort is small, and one possibility would be to wait a few more years to obtain a larger study cohort before publishing and/or gather results from different hospitals (with the same watchful waiting strategy).
Reply 1: Another study is being planned.

Query 2: The simple abstract should be improved. It is not just decrease in anxiety that will be reduced by clinical implementation of biomarker tests for regression. Treatment could also be more individualized with improved prognostication. Women with high risk of progression could be identified earlier.
Reply 2: We have modified the simple abstract as suggested.

Query 3: A table summarizing outcomes for patients with negative vs positive GynTect test in CIN2, CIN3 and CIN2 and 3 combined would be clarifying.
Reply 3: All raw data needed for calculation of the study endpoints are displayed in Figure 1. We developed this figure according to the STARD guideline. For better understanding we followed your proposal and summarized the relationship between methylation and outcome in a bar chart (new Figure 2). Cross tabulation of the underlying data is presented in a new supplementary Table S2.

Query 4: A discussion of other studies investigating potential markers for CIN regression should be included (e.g. PMID: 34830895; PMID: 34226019; PMID: 32332850¸ PMID: 31055567; PMID: 31079058).
Reply 4: We have added a new paragraph in the discussion section and have cited the above studies.

Reviewer 2 Report

    • The study aims to analyze the NPV of the methylation marker panel GynTect for the regression of cervical intraepithelial neoplasia (CIN). The study shows that while the clinical regression was observed for 71% and 56% of women with CIN2 and CIN3, respectively, the study did not observe the significant result of having NPV of >90% against the null hypothesis of NPV <=70%. This study is meaningful for the young women patients diagnosed with CIN to prevent the overtreatment. The manuscript provides a well-written background and detailed methods and acknowledges the limitations of the study. Here are my minor comments.  

  • I would recommend changing the title such as “evaluation of the CIN2/3 lesions regression in CIN2/3 methylation marker-negative patients in a longitudinal study”. The current title does not deliver the aim of the study. 

  • Section 2.2. has technical details and can be shortened. I think line 147-150 describing the cut-off is critical, while other sentences can be provided in the supplementary.  

  • The author provides the overview of the study in Figure 1 along with the result but it would be helpful to provide the figure such as bar plot of the proportion of regression, persistance, and progression by CIN2 and CIN3 patients. This plot would be helpful for the readers to understand the result.  

 Minor editing of English language required

Author Response

Thank you for the valuable comments!

Query 1: I would recommend changing the title such as “evaluation of the CIN2/3 lesions regression in CIN2/3 methylation marker-negative patients in a longitudinal study”. The current title does not deliver the aim of the study. Do you see HPV mixed infections (HPV16+HPV18 or HPV16+HPV52) playing any role in your research.
Reply 1: We have modified the title to “Evaluation of CIN2/3 lesion regression in GynTect® DNA methylation marker-negative patients in a longitudinal study”. We noted only one case with mixed infection and considered this to be irrelevant for further analysis.

Query 2: Section 2.2. has technical details and can be shortened. I think line 147-150 describing the cut-off is critical, while other sentences can be provided in the supplementary.
Reply 2: We would prefer to leave this section as it is.

Query 3: The author provides the overview of the study in Figure 1 along with the result but it would be helpful to provide the figure such as bar plot of the proportion of regression, persistence, and progression by CIN2 and CIN3 patients. This plot would be helpful for the readers to understand the result.
Reply 3: All raw data needed for calculation of the study endpoints are displayed in Figure 1. We developed this figure according to the STARD guideline. For better understanding we followed your proposal and summarized the relationship between methylation and outcome in a bar chart (new Figure 2). Cross tabulation of the underlying data is presented in a new supplementary Table S2. 

Reviewer 3 Report

This is a well -conducted study addressing an important issue in cervical screening that is, the usefulness of the methylation marker detection kit GynTect-Pro as a predictor of regression of high grade cervical disease. A test that could predict disease persistence while also predicting diseased regression could be used in triage to avoid surgical interventions. The cohort is small but sufficiently powered. The results are mainly negative in nature, but it is important to document lack of performance of tests in risk stratification of cervical disease as much as positive performance.

The results are clear and set out in an easy-to-follow manner. Unusually, data from patients who terminated the study, and therefore could not be followed up, were still included in the NPV calculations. However, the rationale behind this approach is well explained and justified in the caveats section of the Discussion.

Due to the small number of GynTect-Pro positive samples, calculating the PPV for disease persistence was compromised, which detracts from the significance of the study. Finally, the finding that HPV16/18 positivity correlated with GynTect-Pro positivity is interesting but a larger study that the one presented here is required to explore this properly.

Table 1 gives details of the cohort but I cannot find any analysis related to the number of live births, contraception, smoker/non-smoker in the results. I think it is probably unethical to record these data if they are not relevant to the study.

Author Response

Thank you for the valuable comments!

Query 1: Table 1 gives details of the cohort but I cannot find any analysis related to the number of live births, contraception, smoker/non-smoker in the results. I think it is probably unethical to record these data if they are not relevant to the study.
Reply 1: Table 1 presents baseline data to describe the study population. Smoking status and parity are known to be associated with spontaneous lesion regression (Koeneman et al. 2019). Taking into account the distribution of this data, we could expect that the lesions of patients in our study should have the potential to regress spontaneously, a crucial assumption for our proof-of-concept study. Our study did not aim at analyzing all prognostic factors. Thus, for clarification, we modified the table description as follows:
“Table S1. Characteristics of CIN2/3 patients for prognostic evaluation of the GynTect® methylation status”